# Seeking the Right Question Towards High-Quality Visual Instruction Generation

## Abstract

Large language models achieve significant improvements in instruction following through training with synthetic data. The self-instruct method generates instructions based on manually selected tasks, establishing an annotation-free paradigm for synthesizing instructions. However, the experience of synthesizing language instructions for LLMs does not directly transfer to visual instruction generation. Visual instructions encompass both images and questions, and questions generated directly from images often struggle to form high-quality instructions. By analyzing real user queries, we summarize the characteristics of high-quality instructions: they require image perception, reasoning, and answerability. We propose a three-stage visual instruction generation pipeline, named "Seeking the Right Question" (SRQ), to produce high-quality instructions. In stage 1, we select 160 instructions that meet high-quality standards as seed questions, categorizing them into eight groups based on multi-modal task types. In stage 2, we introduce capability-driven prompting to generate high-quality questions. In stage 3, we implement an Image Dependency Scoring Mechanism to filter the generated questions. Additionally, we use GPT-4o to directly generate answers, forming <image, question, answer> triples for model training. To demonstrate the effectiveness of SRQ, we construct the high-quality instruction dataset Allava-SRQ from 125,000 images sampled from the Allava dataset. Experimental results show that Allava-SRQ significantly improves the performance of multiple baseline models across various benchmarks. We plan to open-source SRQ and the high-quality instruction dataset Allava-SRQ to promote advancements in the field of visual instruction generation.

## 1 Introduction

Recent advancements in large language models (LLMs) have yielded substantial improvements in instruction-following ability through the utilization of synthetic data (Wang et al., 2022; 2023; Singh et al., 2023; Li et al., 2023c). Among these, self-instruct (Wang et al., 2022) exemplifies the generation of instructions derived from deliberately curated tasks, thereby establishing a paradigm for annotation-free synthetic instruction generation. However, the methodologies employed in synthesizing language instructions for LLMs cannot be directly extrapolated to visual instruction tasks, which inherently encompass both images and associated queries. Typically, queries generated from images present challenges in formulating high-quality instructions. Through an analysis of authentic user inquiries, we identify key characteristics of high-quality instructions: the necessity for image perception, the requirement for reasoning, and the provision of definitive answers. Specifically, high-quality instructions must not rely solely on textual questions devoid of image context, as this would reduce the visual language model (LVLM) to a mere LLM. Furthermore, effective instructions should not merely solicit the identification of image content but should instead necessitate reasoning, as failing to do so risks devolving into basic image captioning tasks. Lastly, the questions posed must be answerable to ensure their utility in training contexts.

Prompting GPT directly to generate questions based on a query image presents several limitations. As illustrated in Figure 2 (left), the generated question, "What's in the picture," resembles a typical image captioning task and fails to enhance the model's ability to follow instructions as effective training data. Figure 2 (middle) demonstrates that the generated question is image-independent;

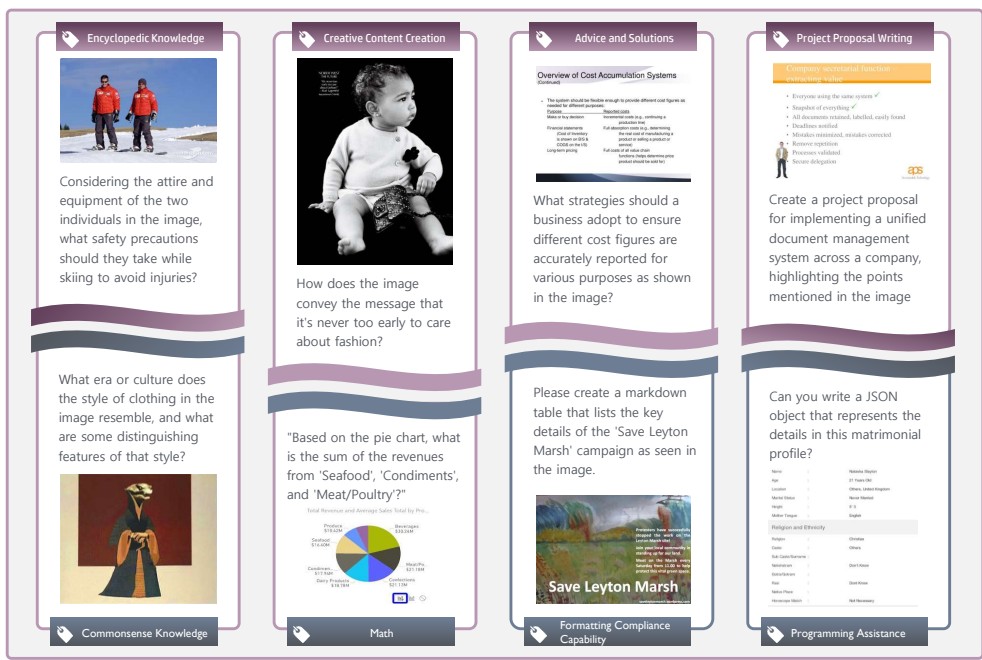

Figure 1: Examples of Generated High-Quality Visual Instructions

thus, the model can answer without perceiving the image, risking the degeneration of the LVLM into a mere LLM. Lastly, as shown in Figure 2 (right), the generated question is unanswerable. While it bears some relevance to the image, it cannot be addressed using the information contained within the image or common knowledge, rendering it ineffective for constructing training data.

Existing methodologies predominantly generate questions based on input images. LLaVA (Liu et al., 2024c) is the first to propose the generation of visual instruction data with the assistance of GPT-4V (Achiam et al., 2023). Following this, Allava (Chen et al., 2024a) employs a caption-then-QA framework, wherein GPT-4V first generates captions for the input image, after which a language model formulates multiple questions based on these captions. Although these methods have proven effective on certain benchmarks, a significant gap remains between the generated questions and those encountered in real-world scenarios. To address this limitation, MMinstruct (Liu et al., 2024d) introduces enhancements by randomly sampling a subset of examples from a curated seed question pool, thereby offering robust prompts for the large language model (LLM) and facilitating the generation of a diverse array of questions. Subsequently, it selects images from a library based on the relevance between the questions and the images to create the final inquiries. However, the questions produced by MMinstruct are susceptible to leaking image content, resulting in some questions being answerable without reference to the images and thereby not fully addressed. Additionally, the entire instruction generation process heavily relies on the capabilities of the LLM, which makes it prone to hallucinations and errors, thus rendering the effective filtering of high-quality instructions a significant challenge.

To address these challenges, we propose a novel visual instruction generation pipeline, termed "Seeking the Right Question" (SRQ), as shown in the 3. In stage 1, we analyze high-quality instructions and select a total of 160 instructions that meet rigorous quality standards as our seed questions. To enhance the diversity and balance of these instructions, we categorize them into eight distinct groups: Project Proposal Writing, Programming Assistance, Mathematics, Formatting Compliance Capability, Encyclopedic Knowledge, Creative Content Creation, Commonsense Knowledge, and Advice and Solutions. In stage 2, we introduce capability-driven prompting to generate high-quality questions. This method utilizes eight categories of seed questions as few-shot examples, prompting GPT-4o (Islam & Moushi, 2024) to simultaneously generate eight different types of questions based on the input image content. Capability-driven prompting not only significantly increases the diversity of command generation but also indirectly reduces the number of input tokens, thereby

Figure 2: Limitations of GPT asking questions directly to images. Left: the generated question, "What's in the picture," resembles a simple image captioning task. Middle: it demonstrates that the generated question is image-independent, and the model can answer without perceiving the image. Right: the generated question is unanswerable.

expediting the generation process. In stage 3, we propose a new visual instruction filtering method, the Image Dependency Scoring Mechanism, which directly evaluates the degree of dependence of the answer to the question on the image content. The highest-scoring question is retained, resulting in the formation of an <image, question> pair. Finally, we employ GPT-4o to directly generate answers, culminating in the creation of <image, question, answer> triples for model training.

Figure 1 presents eight categories of high-quality visual instructions generated through the SRQ method. It is evident that answering these questions relies not only on text comprehension but also on a deep perception and detailed analysis of the image. Each question requires the respondent to extract key information from the image and engage in logical reasoning to arrive at accurate answers. Furthermore, these questions can be answered based solely on the information presented in the image, without the need for additional context or information that is not visible within the scene. These high-quality visual instructions effectively enhance the performance of LVLMs across various tasks.

To demonstrate the effectiveness of SRQ, we construct high-quality instructions from 125,000 images sampled from the Allava dataset. We then conduct extensive experiments on this data, revealing that our approach significantly enhances model performance across various benchmarks. Our dataset yields an improvement of 2.3 points compared to LLAVA and 2.3 points compared to Allava, clearly illustrating the effectiveness of our methodology.

In summary, our contributions are threefold: 1. We propose a novel visual instruction generation method, SRQ, capable of constructing high-quality instructions from images. 2. A high-quality dataset Allava-SRQ is developed based on SRQ, resulting in substantial performance improvements across multiple models after training on this dataset. 3. We will open-source SRQ and the resulting dataset to foster advancements in the field of visual instruction generation.

## 2 METHOD

In this section, we first introduce the selection of seed questions based on the definition of high-quality visual instructions, then we present capability-driven prompting for question generation, and finally, we provide a detailed explanation of the filtering process.

### 2.1 THE SELECTION OF SEED QUESTIONS

Through an analysis of real user queries, we identify key characteristics of high-quality instructions: the necessity for image perception, the requirement for reasoning, and the provision of definitive answers. Specifically, it is imperative that responses to high-quality instructions are not based solely on questions that lack image context, as such an approach would reduce a visual language model (LVLM) to a standard language model (LLM). Furthermore, high-quality instructions must extend beyond simply asking what is depicted in the image and should necessitate reasoning; otherwise, they risk devolving into basic image captioning tasks. Lastly, the questions posed must be answerable; otherwise, they cannot be utilized for training purposes.

Based on our analysis of high-quality instructions, we selected a total of 160 instructions that meet established quality standards as our seed questions. To enhance the diversity and balance of these instructions, we categorize them into eight distinct groups: Project Proposal Writing, Programming Assistance, Mathematics, Formatting Compliance Capability, Encyclopedic Knowledge, Creative

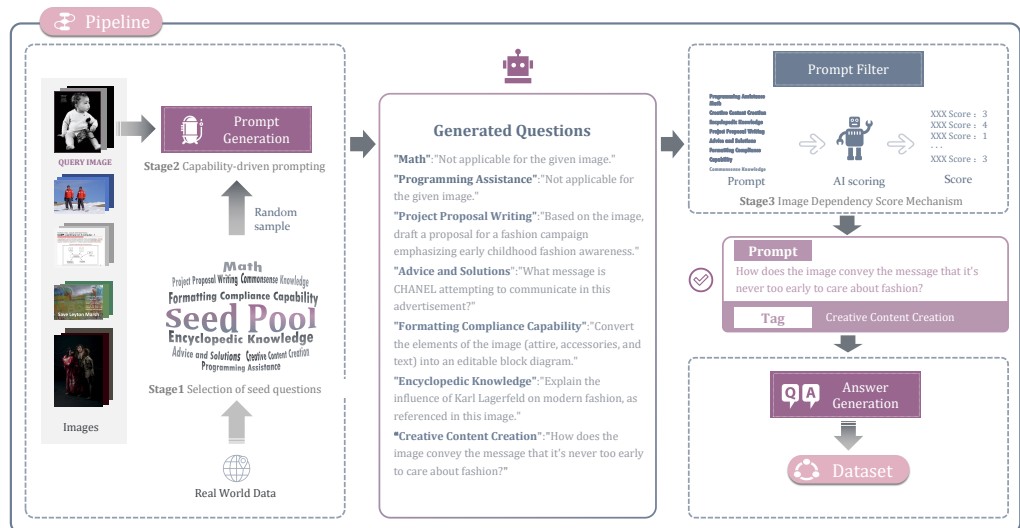

Figure 3: SRQ Pipeline. In stage 1, we select 160 instructions that meet high-quality standards as seed questions, categorizing them into eight groups based on multi-modal task types. In stage 2, we introduce capability-driven prompting to generate high-quality questions. In stage 3, we implement an Image Dependency Scoring Mechanism to filter the generated questions. Additionally, we use GPT-4o to directly generate answers, forming <image, question, answer> triples for model training.

Content Creation, Commonsense Knowledge, and Advice and Solutions. It is noteworthy that we do not classify pure image perception tasks, such as captioning and optical character recognition (OCR), as separate categories, as the ability to accurately answer complex questions relies fundamentally on the model's capacity to perceive images accurately.

In the process of selecting seed questions, it is important not only to assess whether the corresponding instruction meets high-quality criteria but also to ensure that the question content is appropriate. Not all high-quality instructional questions can serve as seed questions. For example, The question in Figure 4(A) is too long and contains a lot of specific information from the original image, which may give people the impression that it is irrelevant to the query image when used for question generation. The question in Figure 4(B) is strongly dependent on the internal logic of the article in the figure, the question pattern is not novel enough, and has insufficient reference value for question generation. In contrast, the question in Figure 4(C) is an answerable general question characterized by complete logical structure. It does not strongly depend on a specific image and avoids excessive reliance on image information, requiring only relevant contextual supplementation based on the content of the query image. A well-formulated seed question can serve as a template for question generation, significantly enhancing the quality of the generated questions.

## 2.2 CAPABILITY-DRIVEN PROMPTING FOR QUESTION GENERATION

As shown in Figure 2, prompting GPT to generate questions based on a query image has several limitations: question valueless, image independence and answer unavailable. As illustrated in Figure 3 (stage2), we propose capability-driven prompting to generate high-quality questions. Different images are suitable for different tasks. Therefore, for each query image, we sample from the eight categories of seed questions outlined in section 2.1 to generate questions across all eight categories simultaneously. To enhance the likelihood of obtaining suitable seed questions for reference, we randomly sample three from each category as few-shot examples.

We prompt GPT-4o using the information contained in the image, refer to the question patterns of the seed questions, and propose questions that necessitate reasoning. The eight categories encompass a diverse range of instruction types to LVLM model. By generating questions across eight categories for each query image instead of limiting to a single category, we significantly increase the

Figure 4: Examples of seed questions. (A) is too long and includes excessive details from the original image. (B) relies heavily on the internal logic of the text in the figure. (C) is an answerable general question.

probability of producing usable questions. This parallel generation approach also indirectly reduces the number of input tokens and accelerates the generation process. Furthermore, employing three seed questions per category as few-shot examples markedly enhances the diversity of the generated questions compared to utilizing just one seed question.

As shown in Figure 3 (generated questions), after capability-driven prompting, GPT-4o generates eight categories of questions for each query image concurrently. In the current query image, which depicts a child and lacks any elements related to mathematics or programming, the contents of the generated questions for these two categories are marked as "Not applicable for the given image." The remaining six categories generate questions based on the image content, guided by the seed questions. These questions undergo scoring and filtering in the stage 3 Image Dependency Scoring Mechanism.

## 2.3 IMAGE DEPENDENCY SCORING MECHANISM

As described in 3 (stage 3), we filter generated questions based on the characteristics of high-quality instructions, which include the necessity for image perception, the requirement for reasoning, and the provision of definitive answers. To achieve this, we propose an Image Dependency Scoring Mechanism. We determine whether the generated questions rely on image perception and whether the answers depend on the image content, scoring the questions accordingly. The specific scoring criteria are as follows:

1. The question cannot be answered based on the information in the image (score = 1).
2. The question can only be roughly answered based on the information in the image, requiring considerable inference (score = 2).
3. The question can be partially answered based on the information in the image, but key details are missing and inference is required (score = 3).
4. The question can be mostly answered based on the information in the image, needing only a few inferred details (score = 4).
5. The question can be completely answered based on the information in the image without inference or assumption (score = 5).

Scores ranging from 1 to 5 represent the quality of the question from low to high.

Regarding the requirement for reasoning in instructions, most questions generated under the guidance of high-quality seed questions necessitate either complex or simple reasoning. Concerning the provision of definitive answers, even the most advanced LVLMs struggle to generate accurate responses for all questions (Tong et al., 2024a). Explicit judgment may lead to hallucinations, making it difficult for the LVLM to assess its ability to answer a question, often resulting in forced responses. Consequently, the assessment of answerability is implicitly included in the scoring mechanism that

Table 1: Overall results. All models utilize CLIP ViT-L/336px as the vision encoder. The * symbol indicates that the vision encoder is trained with LoRA. MMB and SQA refer to MMBench and ScienceQA, respectively. "Ours" refers to the dataset in which we replace the single-round dialogue data in LLava665k with our generated Allava-SRQ.

| Model | Training Data | MMB Test(EN) | MMB Test(CN) | OCR-Bench | SEED-IMG | AI2D Test | SQA Test | Hallusion-Bench aAcc | POPE | MMStar | MMVet |
|---|---|---|---|---|---|---|---|---|---|---|---|
| vicuna1.5-7B | LLava | 66.9 | 60.4 | 33.4 | 64.6 | 58.4 | 67.0 | 46.6 | 87.1 | 33.4 | 32.2 |
| vicuna1.5-7B | Ours | 68.9 | 68.9 | 35.7 | 65.1 | 58.4 | 70.1 | 44.6 | 86.3 | 35.9 | 38.7 |
| vicuna1.5-13B | LLava | 68.9 | 65.3 | 34.1 | 66.4 | 58.3 | 70.6 | 45.1 | 87.5 | 38.2 | 35.9 |
| vicuna1.5-13B | Ours | 69.3 | 65.1 | 37.2 | 67.1 | 60.6 | 71.0 | 45.0 | 86.4 | 35.9 | 42.1 |
| Llama-3-v1.1-8B* | LLava | 72.5 | 69.2 | 37.7 | 70.4 | 60.3 | 71.9 | 46.1 | 86.2 | 38.9 | 35.8 |
| Llama-3-v1.1-8B* | Ours | 73.8 | 70.1 | 42.9 | 70.3 | 62.4 | 74.0 | 49.2 | 87.1 | 42.9 | 40.3 |

relies on image content, offering a more reliable approach than explicit judgments of whether a question is answerable.

## 3 EXPERIMENT

In this section, we demonstrate the effectiveness of our proposed method through qualitative and quantitative experiments. We first introduce our experimental setup, then present the performance of our method on 10 commonly used VLM benchmarks which are MMBench Test(EN), MMBench Test(CN) (Liu et al., 2023b), OCRBench (Liu et al., 2024e), SEEDBench (Li et al., 2023a), AI2D Test (Hiippala et al., 2021), ScienceQA (Saikh et al., 2022), HallusionBench (Guan et al., 2024), POPE (Yifan Li & Wen, 2023), MMStar (Chen et al., 2024b) and MMVet (Yu et al., 2023). Finally, we showcase the design details of our method through point-by-point ablation experiments.

### 3.1 EXPERIMENTAL SETUPS

**Data.** Our fundamental goal is to demonstrate the effectiveness of our proposed visual generation approach, SRQ. To facilitate subsequent experiments, we select two open-source datasets for testing. One is the instruction data proposed by LLava, consisting of 665k samples, which we term as LLava665k. Of these, 100k are single-round dialogue data generated using GPT-4V, termed as LLava100k. Additionally, we randomly sample 125k data from ALLava (Chen et al., 2024a), termed as ALLava125k, for ablation studies.

**Model.** We began with the basic LLaVA-1.5 (Liu et al., 2023a), which utilizes CLIP ViT-L/336px (Radford et al., 2021a) for image encoding and Vicuna v1.5 7B (Zheng et al., 2024) for text encoding. However, this approach does not include any training for the vision encoder, which conflicts with our belief that precise image understanding is crucial for complex reasoning tasks. To address this, we first enhance the baseline. Specifically, we swap the language model for the latest LLaMA 3.1-8B (Vavekanand & Sam) and train the vision encoder with Low-Rank Adaptation(LoRA) (Hu et al., 2021) during the supervised finetune(SFT) phase, while keeping the rest of the training parameters the same.

**Evaluation.** We conduct comprehensive evaluations on 10 vision-language benchmarks as demonstrated in Table 1. These benchmarks cover various question types, including multiple-choice and Q&A, and evaluate the model's abilities from multiple perspectives, such as image perception, mathematics, science, providing a comprehensive assessment of the model's ability.

To provide a clear comparison of the results under different settings, we select MMStar (Chen et al., 2024b), MMVet, OCRBench (Liu et al., 2024e) as representative benchmarks for demonstration in ablation studies. To accurately assess the model's perception capabilities, MMStar (Chen et al., 2024b) manually selected 1,500 questions from six benchmarks—MMBench (Liu et al., 2023b), MathVista (Lu et al., 2023), AI2D (Hiippala et al., 2021), MMMU (Yue et al., 2024), ScienceQA (Saikh et al., 2022), and SeedBench (Li et al., 2023a)—that require visual content to be answered, forming a high-quality and diverse benchmark. MMVet manually constructed 218 challenging questions to evaluate the model's reasoning ability in an open-ended format.

Table 2: Baseline. † represents official implement, the result is obtained from OpenCompass leadboard. * stands for our re-implemented results. Full LLM refers to training all parameters of the LLM during training, Frozen ViT refers to freezing the vision encoder, and LoRA VIT refers to training the vision encoder using the LoRA.

| Model | Fine-tuning Strategy | OCRBench | MMStar | MMVet |
|---|---|---|---|---|
| LLaVA-v1.5-7B † | Full LLM, Frozen ViT | 31.8 | 33.1 | 32.9 |
| LLaVA-v1.5-7B* | Full LLM, Frozen ViT | 32.5 | 33.4 | 32.5 |
| LLaVA-Llama-3-v1.1-8B | Full LLM, Frozen ViT | 32.8 | 39.4 | 32.0 |
| LLaVA-Llama-3-v1.1-8B | Full LLM, LoRA ViT | 37.7 | 39.2 | 35.8 |

**Implementation details.** We employ XTuner (Contributors, 2023) as our training framework. Since LLaMA 3.1 8B is used as the language model, we adjust the batch size per device to 8 and set the accumulation step to 2 to accommodate GPU memory constraints. All other hyperparameters remain aligned with the official open-source configuration. For evaluation purposes, we utilize VLMEval (Duan et al., 2024) as our testing framework.

## 3.2 PERFORMANCE ON VISION-LANGUAGE BENCHMARKS

After generating and filtering visual instructions for the sampled ALLava125k using SRQ, we obtain 100k high-quality visual instruction data, referred to as Allava-SRQ. Figure 8 illustrates the diversity and provides a few examples of Allava-SRQ. We replace the single-round dialogue data, LLava100k, in LLava665k with Allava-SRQ, resulting in a dataset that matches LLava665k in terms of data volume. Keeping all other experimental parameters the same, we compare the performance of the same model trained on different datasets across ten commonly used VLM benchmarks. The experimental results are shown in Table 1. Compared to the standard LLava665k data, the data generated using our SRQ method delivers superior results across multiple benchmarks. Notably, our method demonstrates a significant performance improvement in MMBench, SQA, MMStar, MMVet, and OCRBench, with several models achieving an average gain of 3 points. These benchmarks cover diverse areas, including basic knowledge, scientific understanding, professional skills, and complex reasoning, thereby strongly supporting the effectiveness of our approach.

## 3.3 ABLATION STUDIES

**Baseline.** To thoroughly evaluate the performance of our proposed methods, we first reproduce the accuracy of vanilla LLAVA and then construct a new baseline based on that. The experimental result is recorded in Table 2. By comparing the first two rows, we can see that our re-implemented results in slightly better accuracy than the official LLAVA-provided accuracy. Moreover, after replacing the LLM with LLaMA 3.1 and adding LoRA training to the vision encoder during the SFT stage, the model's performance shows continuous improvement. Comparing the last two rows, we can observe that training the vision encoder during the SFT stage effectively enhances performance on benchmarks like MMVet. This supports the concept that precise image perception contributes to improved performance in complex reasoning tasks. Therefore, unless otherwise stated, the fourth-row model in Table 1 will be used as the baseline model in the rest of the paper.

**Instruction Generation.** To demonstrate the effectiveness of the proposed Capability-driven strategy, we compare it with the most common few-shot prompt, which involves randomly selecting a few samples from the seed pool as examples for question generation. In this experiment, we set the number of few-shot examples to 8. For the same set of images, we use both the capability-driven prompt and the standard few-shot prompt to generate a batch of questions, and evaluate their quality using the scoring method mentioned earlier. Figure 5 shows the frequency distribution histogram of the scores. It can be observed that, compared to our proposed capability-driven prompting, the standard few-shot prompt generated more 1-score questions. Additionally, we manually inspect the few-shot prompts and find that this method tends to first describe the image content in detail before asking the question, as shown in Figure 6. These two points together demonstrate the effectiveness of our proposed method.

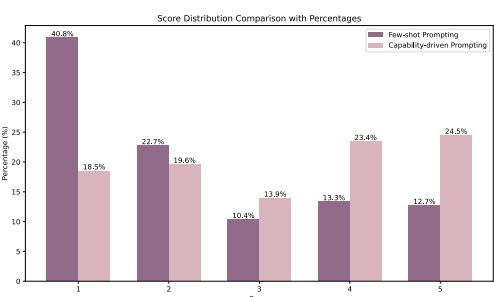

Figure 5: Histogram of the frequency distribution of question scores using different generation methods. Clearly, the quality generated using the standard few-shot prompt is lower.

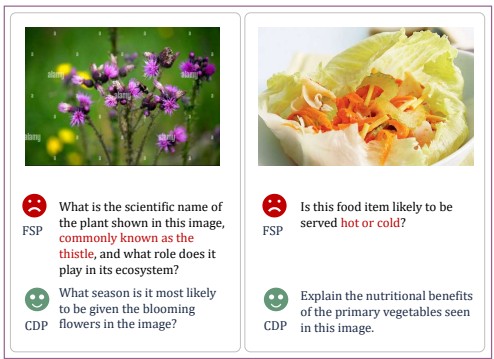

Figure 6: Examples of question generation, where FSP stands for Few-shot Prompting and CDP refers to Capability-driven Prompting.

**Impact of Image Dependency Scoring Mechanism.** To evaluate the effectiveness of the Image Dependency Scoring Mechanism, we divide the data into two groups, a discarded group and a selected group, based on the score threshold. To ensure the fairness of the experiment, we randomly sample data from the larger group to match the size of the other group for testing. The results are presented in Table 6. The second row in the table shows the experimental results of the discarded group with a score threshold of 1. Compared to the baseline, this experiment involves more data but results in a slight performance decline on both MMStar and MMVet. However, when using the selected group, the model's performance slightly improves on OCRBench and MMStar, and achieves a notable gain of 4.6 points on MMVet. This clearly demonstrates the importance of high-quality questions. We then raise the score threshold to 2 and conduct a similar set of experiments. The results for this part are recorded in the last two rows of the table. Compared to the discarded group, the selected group shows noticeable improvements on OCRBench and MMStar, two benchmarks focused on visual perception, with gains of 2 points and 1.6 points, respectively. However, performance remains consistent on MMVet, which emphasizes complex reasoning. This suggests that there is still some high-quality data among questions with a score of 2. Therefore, we ultimately decide to use 1 as the threshold.

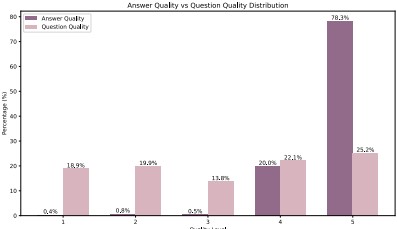

Figure 7: Histogram of frequency distribution for scoring questions and answers on ALLava125k.

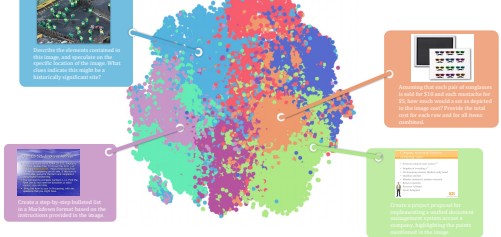

Figure 8: Image distribution of the Allava-SRQ dataset

**Compared with scoring answer.** We follow the scoring method used in the self-reward model (Yuan et al., 2024) and use GPT-4o to score the responses generated by itself. Figure 7 shows the score distribution of questions and answers. We find only about 2% of the responses rated 3 or below. Such an extreme score distribution indicates that the model is not effectively distinguishing low-quality answers, which can lead to inefficient data filtering. To verify this, we design a set of experiments, with the results recorded in Table 4. Since there is too little data with a score below 3, we directly compare the performance of the entire dataset with that of the selected group when the score threshold is set to 4. On average, the performance is almost identical. Then, we increase the score threshold to 5, and compare to the second row where the threshold is 4, we observe varying degrees

Table 3: Qscore is an abbreviation of question score given by the proposed method. "Discarded" refers to the portion of the dataset with a score less than or equal to the score threshold, while "Selected" represents the portion with a score greater than the threshold. We randomly sample data from the selected group to match the size of the corresponding discarded group for fair experiment.

| Extra Image Source | Threshold | Group | OCRBench | MMStar | MMVet |
|---|---|---|---|---|---|
| no extra data(baseline) | - | - | 37.7 | 39.2 | 35.8 |
| ALLava125k | Qscore = 1 | Discarded | 40.8 | 38.6 | 35.4 |
| ALLava125k | Qscore = 1 | Selected | 41.1 | 39.1 | 40.0 |
| ALLava125k | Qscore = 2 | Discarded | 40.9 | 39.1 | 39.2 |
| ALLava125k | Qscore = 2 | Selected | 42.9 | 40.7 | 39.2 |

Table 4: Qscore and Ascore are abbreviations of question score and answer score respectively. "Selected" represents the portion with a score greater than the threshold.

| Extra Image Source | Threshold | Group | OCRBench | MMStar | MMVet |
|---|---|---|---|---|---|
| no extra data(baseline) | - | - | 37.7 | 39.1 | 35.8 |
| ALLava125k | Ascore =1 | Selected | 43.4 | 41.5 | 39.7 |
| ALLava125k | Ascore = 4 | Selected | 43.5 | 40.9 | 41.0 |
| ALLava125k | Ascore = 5 | Selected | 42.0 | 40.3 | 40.6 |
| ALLava125k | Qscore = 2 | Selected | 42.2 | 42.3 | 42.6 |

of performance decline across the three benchmarks. This suggests that there is still a significant amount of high-quality data in the group with an answer score of 4, which strongly confirms that directly scoring answers is both challenging and inefficient. Next, we conduct a direct comparison between question scoring and answer scoring. We randomly sample data from the portion with a question score of 2 or higher, matching the quantity of data with an answer score of 5, and ran the experiment. The results are recorded in the last row of the table. Compared to the method of directly scoring answers, our proposed method achieve a significant 2-point improvement on both MMStar and MMVet, indicating that our approach is more efficient at filtering.

**Compared with other visual instruction generation approach.** To enable a direct comparison with other visual instruction generation approaches, we modify the corresponding instructions while keeping the image content consistent, and then train the models. We first remove the non-conversational instructions generated by GPT-4o in LLAVA, which amounts to 100k instructions. Then we use SRQ to construct a new set of instructions for these 100k images. To maintain consistency in data volume, for each image, we select the question with the highest filter score (even if the highest score is 1) for subsequent answer generation. The first two rows of Table 5 show the results of this experiment. It can be observed that when using SRQ for generation, the model's performance increases significantly, with a 3.3-point improvement on MMVet and a 3.6-point improvement on MMStar. This clearly demonstrates the effectiveness of the generation method. Next, we use SRQ to generate instructions, randomly sample 100k data points, and then replace the instructions with those generated by Allava, creating a comparative experiment. The last two rows of the table show the results of this experiment. Compared to Allava, our dataset significantly improves the model's performance on complex tasks, and also shows gains on MMStar, which tests perception abilities. This clearly demonstrates the importance of generating challenging questions and the effectiveness of SRQ.

## 4 RELATED WORKS

**Large Vision-Language Model.** LLMs have driven significant advancements in artificial intelligence, and LVLMs have emerged as a key area of research owing to their extensive potential for real-world applications. Vision language models demonstrated by CLIP (Radford et al., 2021b) and subsequent works (Fang et al., 2023; Jia et al., 2021; Li et al., 2022; 2023b; Sun et al., 2024; Zhang et al., 2022) facilitate the confluence of visual and textual modalities through contrastive learning. LLaVA (Liu et al., 2024b) innovatively leverage vision transformer-based CLIP models and par-

Table 5: Compared with other visual instruction generation approaches, * indicates that no images will be discarded by the filter. Avg. stands for the average score across three benchmarks.

| Image Source | Generation method | OCRBench | MMStar | MMVet | Avg. |
|---|---|---|---|---|---|
| LLava100k | LLava | 37.7 | 39.1 | 35.8 | 37.7 |
| LLava100k | SRQ* | 39.3 | 41.7 | 39.1 | 40.0 |
| ALLava125k | ALLava | 40.6 | 41.7 | 36.7 | 39.7 |
| ALLava125k | SRQ | 42.9 | 42.9 | 40.3 | 42.0 |

tially replicate the capabilities of GPT-4V. Numerous models (Chen et al., 2023a;b; Bai et al., 2023; Peng et al., 2023; Ye et al., 2023; Lu et al., 2024) emerge in succession subsequently and enhance capabilities through optimizing pre-training and fine-tuning instruction data or modifying model architectures. Heavy works (Dai et al., 2023; Liu et al., 2024a; McKinzie et al., 2024) underscores the importance of high-quality instruction data in enhancing model performance. Diverse image sources augment perceptual capabilities, while well-crafted questions enhance reasoning. However, high-quality visual instructions are scarce and their manual construction is costly. To address this, we propose SRQ, an automated approach to generate high-quality visual instructions from images.

**Multimodal Data Construction.** High-quality visual instruction data is an essential component while training LVLMs. In the field of LLMs, Self-Instruct (Wang et al., 2022) introduces a semi-automated process, enabling the generation of arbitrarily large datasets. However, visual instruction data necessitates the alignment of language directives with image content. InstructBLIP (Dai et al., 2023) converts existing visual-language datasets into an instruction-tuned format. LLaVA (Liu et al., 2024b), VisionLLM (Wang et al., 2024), and Shikra (Chen et al., 2023b) employ GPT prompts during data generation, but these approaches are often constrained by whether the dataset includes captions or bounding boxes, limiting their broad applicability. ShareGPT4V (Chen et al., 2023c) utilizes GPT-4V to generate high-quality image captions and expands the dataset to 1.2 million examples. ALLava (Chen et al., 2024a) leverages GPT-4V's ability to generate detailed captions, complex reasoning instructions, and in-depth image-based answers to create a synthetic dataset. Genixer (Zhao et al., 2023) further investigated whether self-instruction could be achieved without relying on GPT-4V's capabilities MM-Instruct (Liu et al., 2024d) innovatively uses ChatGPT to automatically generate diverse instructions from a small set of seed instructions. Despite significant efforts in data construction, a fully automated pipeline that can generate and evaluate high-quality data simulating real-world conditions has yet to be realized.

**LLM-as-a-Judge.** The use of "LLM-as-a-Judge" prompts has become a common method for evaluating language models, with benchmarks like LLavaBench (Liu et al., 2023a) and MMVet (Yu et al., 2023) relying on LLMs for scoring. Self-Rewarding Language Models (Yuan et al., 2024) advance this by incorporating self-scoring mechanisms as a reward model to aid in training. However, some studies (Huang et al., 2023) have pointed out that LLMs still struggle with consistent self-correction. This problem is even more pronounced in vision-language models, where even the most advanced systems often fail at image recognition tasks that humans find trivial (Tong et al., 2024b). This raises further concerns about the reliability of using large models as evaluators in visual instruction tasks.

## 5 CONCLUSION

In this paper, we propose Seeking the Right Question(SRQ), a method for automatically generating high-quality visual instructions based on the content of input images. This approach utilizes Capability-Driven Prompting for Question Generation to ask questions about the image from multiple perspectives. It then applies the Image Dependency Scoring Mechanism to directly filter the generated questions, resulting in a set of high-quality visual instruction data. Experiments have shown that training various models on this dataset significantly improves their performance, demonstrating the effectiveness of our method.

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

# A APPENDIX

## Capability-Driven Prompting For Question Generation

As an expert with extensive knowledge in various disciplines, you possess a profound understanding of the information content within images and know how to formulate questions that require a certain level of reasoning and utilize the information contained in the images. I will provide you with 8 dimensions and an image; these dimensions are for your reference when formulating questions about the image. Your questions must incorporate at least one of these dimensions. First, I will provide you with the list of the 8 dimensions and example questions for each dimension as follows:
"Capability : Commonsense Knowledge" (Three Examples)
"Capability : Encyclopedic Knowledge" (Three Examples)
"Capability : Math" (Three Examples)
"Capability : Project Proposal Writing" (Three Examples)
"Capability : Programming Assistance" (Three Examples)
"Capability : Creative Content Creation" (Three Examples)
"Capability : Advice and Solutions" (Three Examples)
"Capability : Formatting Compliance Capability" (Three Examples)
Please organize the response in JSON format, ensuring that your answer strictly follows the format below:

```
Output:{ "Math":  "Question that meets the requirement",
  "Programming Assistance":  "Question that meets the
requirement",
  "Creative Content Creation":  "Question that meets the
requirement",
  "Commonsense Knowledge":  "Question that meets the
requirement",
  "Project Proposal Writing":  "Question that meets the
requirement",
  "Advice and Solutions":  "Question that meets the
requirement",
  "Formatting Compliance Capability":  "Question that meets
the requirement",
  "Encyclopedic Knowledge":  "Question that meets the
requirement" }
```
The above are some example questions that include the corresponding dimensions.

## Image Dependency Scoring Mechanism

You are an evaluator tasked with rating the quality of a question based on an image provided. Your goal is to give a score from 1 to 5, reflecting how well the question can be answered using the image. Here's the scoring guide:
1: The question cannot be answered based on the information from the image.
2: The question can only be loosely answered based on the information from the image, with significant inference required.
3: The question can be partially answered based on the information from the image, but key details are missing and require inference.
4: The question can mostly be answered based on the information from the image, with only minor details requiring inference.
5: The question can be fully answered based on the information from the image, with no inference or assumptions required.
Please evaluate the following:
Image:
Question:
Please first briefly describe your reasoning (in less than 100 words), and then write "Score: " in the last line.

Table 6: A set of experiments using Qwen2-VL-72B. G-Model is an abbreviation for Generation Model. GQQ stands for question generation by GPT-4o, question quality bu Qwen and Answer Generation by Qwen. QQQ stands for all the three stages generated by Qwen.

| Extra Image Source | Threshold | G-Model | Group | OCRBench | MMStar | MMVet |
|---|---|---|---|---|---|---|
| no extra data(baseline) | | - | - | 37.7 | 39.2 | 35.8 |
| ALLava125k | Qscore = 1 | GQQ | Discarded | 40.1 | 37.5 | 34.1 |
| ALLava125k | Qscore = 1 | GQQ | Selected | 40.3 | 39.7 | 36.1 |
| ALLava125k | Qscore = 1 | QQQ | Discarded | 39.3 | 38.0 | 35.0 |
| ALLava125k | Qscore = 1 | QQQ | Selected | 41.4 | 38.8 | 35.7 |

**Prompt For Scoring Answer**

Here is a question which contains an image and a corresponding instruction from an user and a candidate response. Please grade the response on a 5-point scale using the following criteria:

1: It means the answer is incomplete, vague, off-topic, controversial, or not exactly what the user asked for. For example, some content seems missing, numbered list does not start from the beginning, the opening sentence repeats user's question. Or the response is from another person's perspective with their personal experience (e.g. taken from blog posts), or looks like an answer from a forum. Or it contains promotional text, navigation text, or other irrelevant information.

2: It means the answer addresses most of the asks from the user. It does not directly address the user's question. For example, it only provides a high-level methodology instead of the exact solution to user's question.

3: It means the answer is helpful but not written by an AI Assistant. It addresses all the basic asks from the user. It is complete and self contained with the drawback that the response is not written from an AI assistant's perspective, but from other people's perspective. The content looks like an excerpt from a blog post, web page, or web search results. For example, it contains personal experience or opinion, mentions comments section, or share on social media, etc.

4: It means the answer is written from an AI assistant's perspective with a clear focus of addressing the instruction. It provide a complete, clear, and comprehensive response to user's question or instruction without missing or irrelevant information. It is well organized, self-contained, and written in a helpful tone. It has minor room for improvement, e.g. more concise and focused.

5: It means it is a perfect answer from an AI Assistant. It has a clear focus on being a helpful AI Assistant, where the response looks like intentionally written to address the user's question or instruction without any irrelevant sentences. The answer provides high quality content, demonstrating expert knowledge in the area, is very well written, logical, easy-to-follow, engaging and insightful.

Please evaluate the following:

Image:

Question:

Answer:

Please first briefly describe your reasoning (in less than 100 words), and then write "Score: " in the last line.

