# OpenReview forum: "Seeking the Right Question: Towards High-Quality Visual Instruction Generation"
_ICLR.cc/2025/Conference — Submitted to ICLR 2025_

### Official Review · Reviewer_MRcm · 2024-10-30

**Soundness:** 2
**Presentation:** 3
**Contribution:** 2
**Rating:** 3
**Confidence:** 5

**Summary:**

The author believes that traditional methods struggle to create high-quality visual instructions because they often rely on questions generated directly from images, which may not effectively challenge the models to improve their reasoning or perception skills. The paper introduces a new visual instruction generation method, "Seeking the Right Question" (SRQ). The SRQ pipeline consists of three stages. In stage 1, high-quality seed questions are selected, meeting standards of image perception, reasoning, and answerability. These questions are then categorized into eight groups, such as Mathematics and Creative Content Creation. In stage 2, capability-driven prompting is employed to generate diverse questions using these categories as prompts, leveraging the power of GPT-4o. In stage 3, an Image Dependency Scoring Mechanism is used to filter questions, ensuring they depend heavily on image content, resulting in effective <image, question, answer> triples for model training. The effectiveness of SRQ is demonstrated by constructing a dataset, Allava-SRQ, from 125,000 images, showing significant performance improvements over previous methods like LLAVA and Allava.

**Strengths:**

The paper presents its concepts and findings with considerable clarity, making it easy for readers to follow the arguments and grasp the significance of the work. The authors clearly define the problem, specifically addressing the mismatch issues between questions and images in current visual instruction datasets, which is a well-recognized challenge in the field. Additionally, the approach introduced in the paper is straightforward and intuitive, making it easier for other researchers or practitioners to implement and build upon.

**Weaknesses:**

(1) Overclaim: "The eight categories encompass nearly all task scenarios applicable to the LVLM model." I completely disagree with the author’s statement. These eight categories do not cover all task scenarios and some classifications are not typically applicable to the LVLM model. Please provide further widely accepted theory or experimental evidence, rather than arbitrary assumptions.

(2) Scoring is unreasonable: Many real-world problems involve a coupling of multiple sub-tasks. Some basic sub-problems fit "The question can be completely answered based on the information in the image without inference or assumption (score = 5)," but subsequent sub-questions may require additional reasoning. Using such a standard for filtering can lead to questions that are too simple and do not match the needs of real-world users. Similarly, the subsequent reasoning is flawed: "Consequently, the assessment of answerability is implicitly included in the scoring mechanism that relies on image content, offering a more reliable approach than explicit judgments of whether a question is answerable."

(3) The method is inefficient and difficult to scale: The author's pipeline essentially combines synthesis and filtering. If there is enough original instruction data, using a conventional synthesis method and then filtering to retain high-scoring data could achieve the same effect. Moreover, the author's approach requires generating 8 different questions per image. If a baseline method allows the VLM to randomly generate 8 times more data and then uses a filtering strategy, it might achieve similar or even better results. Since your eight types of questions are completely different, as shown in your case, there is necessarily redundancy. From this perspective, letting the model generate by itself might be better. Thus, important baselines/ablation experiments are missing here.

(4) The method is limited to improving quality at the case level and doesn't ensure or enhance diversity at the dataset distribution level, which is also very important. The authors set eight different categories and 160 seed samples, but there are no additional restrictions on the distribution of input images. If there are many similar images in the vast source collection, how can they ensure the diversity of the final synthesized dataset?

(5) I believe the authors have underestimated the capabilities of VLM, especially GPT-4o. Consolidating the pipeline into a dynamic prompt (including CAPABILITY-DRIVEN PROMPTING FOR QUESTION GENERATION and IMAGE DEPENDENCY SCORING MECHANISM) and integrating it into GPT-4o for synthesizing questions and answers should be more efficient. Therefore, strong experimental comparisons are needed to demonstrate that the effectiveness gained by sacrificing efficiency is significant.

(6) The complete pipeline prompt template is not provided, making it difficult to verify the method's effectiveness at the case level.

**Questions:**

(1) In the experiments, does the version of GPT-4o remain consistent with the model version used in the baseline? There are capability differences between various 4o versions.
(2) As VLMs become increasingly powerful, will the benefits of the proposed solution diminish?
(3) Has there been any testing on the effectiveness of open-source model synthesis?

---

> ### Author Response · Authors · 2024-11-27
>
> Thank you for your thoughtful review and helpful advice.  For weakness 1 and 6 that you mentioned, we will address them in the revision by updating the phrasing to "encompass a diverse range of instruction types" and including the specific prompt templates in the appendix.
>
> For weakness 2: The issue of multi-turn dialogue data synthesis that you mentioned is indeed an interesting topic. However, our pipeline design does not specifically address this aspect, and in the experiments, we retained LLAVA's multi-turn data while only replacing the single-turn data for testing purposes. The proposed pipeline is capable of synthesizing multi-turn data, we leave this as future work.
>
> For weakness 3: One of the primary motivations for proposing an automated synthesis pipeline is to address the current shortage of VLM data. Ensuring the quality of generated data requires a combination of generation and filtering stages. While generating 8 questions may introduce some redundancy, certain combinations can produce unexpected and valuable outcomes as shown in Figure 1. Such surprising questions are difficult to achieve through repeated generation. Moreover, randomly generating 8 times is not more efficient than generating 8 diverse question types simultaneously.
>
> For weakness 4: We agree that the diversity of images significantly impacts the quality of the synthesized data. However, this paper focuses on how to generate and filter higher-quality questions given a specific set of input images. This is why we titled the paper 'Seeking the Right Question.' Additionally, determining in advance which images are suitable for generating questions and what types of questions are appropriate for those images is inherently challenging. For example, vastly different image categories, such as shoes and houses, can both be relevant to advertisement-related questions. As a result, while the images may appear visually diverse, the generated questions might still converge toward a specific type. Therefore, we believe that obtaining diverse and high-quality image sources is another critical topic in the field of data synthesis.
>
> For weakness 5, combining the two stages into a single phase is feasible, and we have conducted experiments to test this approach. The overall performance was slightly lower compared to the split-stage setup. Additionally, integrating the stages makes it more challenging to pinpoint issues and optimize performance. For these reasons, we ultimately decided to present the two stages separately.
>
> For question 1: The version of GPT-4o we use is gpt-4o-2024-05-13.
>
> For question 2: On the contrary, we believe that as model capabilities improve, the effectiveness of this pipeline will further increase. Enhanced models can generate higher-quality questions and produce more accurate and refined answers, thereby elevating the overall performance of the pipeline.”
>
> For question 3: The following presents a set of experiments conducted using Qwen2-vl-72B. The first set of experiments replaces both the scoring and answering models with Qwen, while the second set replaces all models in the pipeline with Qwen. The score threshold here is 1. Q-Gen, Q-Quality, and A-Gen are abbreviations for Question Generation, Question Quality, and Answer Generation, respectively. The results show that the Selected group outperforms the Discarded group and both exceed the baseline performance, aligning with the conclusions in Table 3. These findings demonstrate that our pipeline continues to achieve performance improvements even when using open-source models.
>
> | Group     | Q-Gen | Q-Quality | A-Gen | MMStar | OCRBench  | MMVet |
> |-----------|-------|-----------|-------|--------|------|-------|
> | baseline  | -     | -         | -     | 39.2   | 37.7 | 35.8  |
> | Discarded    | GPT-4o   | Qwen      | Qwen  | 40.1   | 37.4 | 34.1  |
> | Selected      | GPT-4o   | Qwen      | Qwen  | 40.3   | 39.7 | 36.1  |
> | Discarded       | Qwen  | Qwen      | Qwen  | 39.3   | 38.0 | 35.0  |
> | Selected      | Qwen  | Qwen      | Qwen  | 41.4   | 38.8 | 35.7  |
>
>
> We will include the relevant experimental results in the revision. We sincerely appreciate your detailed and thoughtful feedback. If there are any points that remain unclear, we would be happy to discuss them further with you.

---

### Official Review · Reviewer_h17W · 2024-11-01

**Soundness:** 2
**Presentation:** 2
**Contribution:** 2
**Rating:** 3
**Confidence:** 5

**Summary:**

This paper proposes "Seeking the Right Question" (SRQ), a novel three-stage pipeline for high-quality visual instructions generation.
Based on this data pipeline, authors construct the Allava-SRQ dataset based on the 125K images samples from Allava. Experimental results show that the model trained with Allava-SRQ performs better than the counterpart trained with the dataset proposed in LLaVA, demonstrating the effectiveness of the proposed dataset.

**Strengths:**

1. This paper concentrates on the quality of questions in the dataset, which is a crucial component determining the overall quality of the dataset.

2. The proposed data construction pipeline is straightforward.

3. Experimental results show that the model trained on Allava-SRQ outperforms its counterpart trained on the LLaVA dataset, demonstrating the effectiveness of the proposed dataset.

**Weaknesses:**

1. The writing in this paper is lacking clarity. The authors do not adequately explain the motivation behind the design of the proposed data construction pipeline. Additionally, the description of the image dependency scoring mechanism is confusing—how is the score derived for a given image or question?

2. This paper focuses on question quality. However, aside from Figure 5, it does not provide additional quantitative analyses regarding the question quality of other datasets.

3. Since questions are generated using 160 seed instructions, the diversity of generated instructions may be limited.

4. The improvements over the baseline appear to stem from the distillation of GPT-4. The effectiveness of the proposed data pipeline remains unclear.

**Questions:**

Please see weakness.

---

> ### Author Response · Authors · 2024-11-27
>
> Thank you for your thoughtful review. The primary goal of this paper is to propose an automated data synthesis pipeline to enhance the instruction-following capabilities of visual language models (VLMs). To achieve this, we focus on two key aspects: question generation and question filtering, introducing methods to ensure both the diversity and quality of the generated data. In the experiments, we demonstrate the impact of high-quality data on model performance by keeping the data volume and hyperparameters consistent across comparisons.
>
> Table 5 presents the results of our method compared to other data synthesis approaches, showing that our approach significantly improves model performance, which highlights the higher quality of the data we generate. Additionally, we evaluate the diversity of our data through experimental results. These benchmarks cover a variety of question types, including multiple-choice and Q&A, and assess the model's capabilities across multiple dimensions such as image perception, mathematics, and science, providing a comprehensive evaluation of its abilities. The results indicate that models trained on data generated by our method exhibit performance improvements across multiple benchmarks, further validating the diversity of the generated questions.
>
> To demonstrate the effectiveness of our method, we conduct a set of experiments using Qwen2-vl-72B. The first set of experiments replaces both the scoring and answering models with Qwen, while the second set replaces all models in the pipeline with Qwen. The score threshold here is 1. Q-Gen, Q-Quality, and A-Gen are abbreviations for Question Generation, Question Quality, and Answer Generation, respectively. The results show that the Selected group outperforms the Discarded group and both exceed the baseline performance, aligning with the conclusions in Table 3. These findings demonstrate that our pipeline continues to achieve performance improvements even when using open-source models.
>
> | Group     | Q-Gen | Q-Quality | A-Gen | MMStar | OCRBench  | MMVet |
> |-----------|-------|-----------|-------|--------|------|-------|
> | baseline  | -     | -         | -     | 39.2   | 37.7 | 35.8  |
> | Discarded    | GPT-4o   | Qwen      | Qwen  | 40.1   | 37.4 | 34.1  |
> | Selected      | GPT-4o   | Qwen      | Qwen  | 40.3   | 39.7 | 36.1  |
> | Discarded       | Qwen  | Qwen      | Qwen  | 39.3   | 38.0 | 35.0  |
> | Selected      | Qwen  | Qwen      | Qwen  | 41.4   | 38.8 | 35.7  |
>
> Lastly, most existing data synthesis methods rely on more powerful models for data generation, we would greatly appreciate any specific suggestions that could help us further address your concern about the effectiveness of the proposed data pipeline effectively.

---

> > ### Comment · Reviewer_h17W · 2024-11-27
> >
> > Thank you for your response. My primary concern with this paper lies in understanding why the data pipeline was designed in its current form. Clarifying the key insights behind this design, discussing the limitations of previous pipelines, and explaining how your design choices overcome these issues are crucial for helping readers better comprehend your paper.
> >
> > Beyond the writing and explanations, a more significant concern is whether the proposed method is as effective as claimed. As shown in the experimental results in Table 5, the improvements appear marginal, and the base model used is not particularly strong. This raises concerns about its effectiveness when applied to current state-of-the-art models. For instance, MiniCPM-V-2.6-8B achieves scores of 85.2, 57.5, and 60.0 on OCRBench, MMStar, and MMVet, respectively, significantly outperforming the fine-tuned model used in this study.
> >
> > I would consider raising my rating if you could demonstrate the effectiveness of your method on leading models. Otherwise, I will maintain my negative recommendation.

---

> > > ### Author Response · Authors · 2024-12-01
> > >
> > > Thank you for your response. We would like to clarify the fundamental objective of this paper. Rather than pursuing state-of-the-art (SOTA) model performance, our research focuses on developing a novel method for high-quality data synthesis. We demonstrate the effectiveness of our method by showing that, **with the same data volume, our synthesized data leads to superior model performance compared to existing approaches.**
> > >
> > > The core of our work lies in addressing a significant challenge: automated synthesis of high-quality question-VLM (Visual Language Model) data. To tackle this, we developed the SRQ (Synthetic Question Generation) pipeline. Our extensive experimental results demonstrate two key findings: first, given equal data volume, higher-quality data consistently yields superior performance compared to lower-quality alternatives; second, our SRQ pipeline outperforms existing data synthesis methods.
> > >
> > > Regarding the comparison with MiniCPM-V-2.6-8B, the disparity in training data volume makes direct comparisons inappropriate. MiniCPM-V-2.6-8B utilizes 200 million pre-training samples and over 2 million supervised fine-tuning (SFT) samples, whereas LLaVA employs 558,000 and 665,000 samples respectively. Such substantial differences in data volume, along with variations in training configurations, naturally lead to performance disparities.
> > >
> > > We chose LLaVA as our baseline for two strategic reasons. First, its streamlined architecture allows us to effectively isolate and analyze the impact of data quality while maintaining consistent training volumes. Second, LLaVA's significant impact is evidenced by its over 4,000 academic citations and 20,500 GitHub stars, making it a widely recognized benchmark in both academic and industrial contexts. This widespread adoption strengthens the relevance and applicability of our findings.

---

> > > > ### Comment · Reviewer_h17W · 2024-12-02
> > > >
> > > > Thank you for your response.
> > > > MLLMs have made tremendous progress over the past year. Despite the significant impact of the LLaVA series, the performance of LLaVA-1.5 is now noticeably behind compared to more recent models.
> > > > Considering that MiniCPM-V has not open-sourced their training data, a straightforward alternative is to validate the effectiveness of the proposed method on LLaVA-OneVision, which has open-sourced their training data.
> > > > At the same time, given that the improvement of the proposed method in the LLaVA-1.5 setting is marginal, I hold a pessimistic view regarding its ability to maintain effectiveness in a stronger setting. I believe that validating the effectiveness of the proposed method in a state-of-the-art setting is crucial, as it directly determines whether the proposed method can further advance the MLLM field.
> > > > Without such validation, it remains uncertain whether the method is truly capable of advancing the field or if it is simply effective within more limited or outdated contexts.

---

### Official Review · Reviewer_bcb8 · 2024-11-04

**Soundness:** 3
**Presentation:** 3
**Contribution:** 3
**Rating:** 6
**Confidence:** 3

**Summary:**

This paper proposes a new method for generating visual instruction data with better quality, which leads to better performance of downstream VLM trained with the generated data.

**Strengths:**

The quality of synthetic dataset is important in fine-tuning popular visual language models nowadays. The proposed method may effectively improve the quality of generated data, leading to better performance of the resulting visual language models.

Comprehensive experiments are conducted, showing that the proposed method is indeed effective, and outperforms baseline methods.

In experiments, the authors try to make sure that all the compared methods are evaluated under the same setting, by using the same scale of dataset and using carefully chosen fine-tuning strategy, which is important for fair comparison.

**Weaknesses:**

Directly using the scoring for comparison as in Figure 5 may not be appropriate. The scoring criteria described in section 2.3 does not cover all the three aspects the authors mentioned, it may lean towards questions which can be easily answered from the image without reasoning. To compare the quality, we may need human evaluation or better scoring strategy.


The authors mentioned that "for each query image, we sample from the eight categories ... to generate questions across all eight categories simultaneously. ... we randomly sample three from each category as few-shot examples". While for baseline methods, the authors "set the number of few-shot examples to 8. If I understand correctly, the baseline method should also be provided more few-shot (24) samples, to make sure that the generation are compared fairly.


A cost/computation comparison between the proposed method and baseline generation method are needed. The proposed instruction generation may need more computation cost because it may need both generation and scoring. What will the result be if we also apply scoring and filtering in vanilla instruction generation?

**Questions:**

If we use different or ensemble of language models as evaluator for scoring, will it lead to better results?

---

> ### Author Response · Authors · 2024-11-27
>
> Thank you for your detailed comments. Below, we address your concerns point by point:
>
> Regarding the review of generation quality: We conducted manual evaluations to compare the outputs of different generation methods. As mentioned in the original text, “Additionally, we manually inspect the few-shot prompts and find that this method tends to first describe the image content in detail before asking the question, as shown in Figure 6,” this conclusion was drawn from reviewing over 100 samples. However, performing manual evaluations on a statistically significant number of samples is prohibitively costly. Thus we also use score distribution in Figure 5 to show the quality of generation. We believe this hybrid approach balances quality and practicality.
>
> Regarding the number of few-shot examples:
> In the standard few-shot setup, the model is provided with 8 examples as input. Increasing this to 24 examples significantly slows down the generation process. Moreover, we conducted manual evaluations and observed no substantial improvement in the diversity or quality of generated questions when increasing the number of examples. Thus, we keep this part simple and efficient.
>
> Regarding the cost of the pipeline:
> The cost of a language model primarily depends on the number of input and output tokens. While our pipeline includes both question generation and an additional scoring step, we significantly reduce the input token count by using only the question portion from the seed instructions in the few-shot setup. Additionally, the scoring step produces short and simple outputs, resulting in minimal additional output tokens. In practice, our method is faster than the standard few-shot approach.
>
> Regarding the use of ensemble methods:
> Applying ensemble methods to the question quality scoring component alone may not yield significant improvements, as the quality of the answers during supervised fine-tuning (SFT) training is equally critical. Performance improvements might be achievable if ensembles were also applied during answer generation. To investigate the impact of the model, we conduct a set of experiments using Qwen2-vl-72B. The first set of experiments replaces both the scoring and answering models with Qwen, while the second set replaces all models in the pipeline with Qwen. The score threshold here is 1. Q-Gen, Q-Quality, and A-Gen are abbreviations for Question Generation, Question Quality, and Answer Generation, respectively. The results show that the Selected group outperforms the Discarded group and both exceed the baseline performance, aligning with the conclusions in Table 3. These findings demonstrate that our pipeline continues to achieve performance improvements even when using open-source models.
>
> | Group     | Q-Gen | Q-Quality | A-Gen | MMStar | OCRBench  | MMVet |
> |-----------|-------|-----------|-------|--------|------|-------|
> | baseline  | -     | -         | -     | 39.2   | 37.7 | 35.8  |
> | Discarded    | GPT-4o   | Qwen      | Qwen  | 40.1   | 37.4 | 34.1  |
> | Selected      | GPT-4o   | Qwen      | Qwen  | 40.3   | 39.7 | 36.1  |
> | Discarded       | Qwen  | Qwen      | Qwen  | 39.3   | 38.0 | 35.0  |
> | Selected      | Qwen  | Qwen      | Qwen  | 41.4   | 38.8 | 35.7  |
>
> Thank you for your suggestions. If there are any points that remain unclear, we would be happy to discuss them further with you.

---

### Meta-Review · Area_Chair_HmJy · 2024-12-19

**Metareview:**

This work proposed to synthesize high-quality visual question answering pairs for training multimodal LLMs. The authors executed the data synthesis in three steps: 1) select seed questions; 2) generate high-quality questions from seeds; and 3) filter generated questions and use GPT-4o to get the answers. Using this data, the trained model outperforms the baselines as shown in the experiments.

The main contribution of this work is proposing a new dataset for multimodal LLM training, and the authors promised to release the training data. Nevertheless, as pointed out by reviewers, one prominent concern is that the improvement brought by the new dataset is marignal across the board and there is no clear analysis or explanation from the authors on why the performance should be improved. Another main concern shared by the reviewers as well as the ACs is the scalability of the proposed method. The main contributor to the proposed dataset is GPT-4o. As such, people can hardly follow the same routine to scale up the training data. Moreoever, it is not clear how to further enrich the diversity or coverage of different image and text domains for the dataset.

Given that this work recieves two rejects and one marginal accept, the ACs think that it is not ready to be published at this venue, and encourage the authors to further polish this work from different aspects, including more thorough experimental analysis, better baseline methods and more systematical pipeline.

**Additional Comments On Reviewer Discussion:**

The authors' rebuttal addressed the concerns raised by the reviewers to some extent. Unfortunately, only one reviewer (h17W) responded to authors' response but keep the rating with most of the concerns still unsolved. The ACs read the discussions carefully and agree with the reviewer.

---

### Decision · Program_Chairs · 2025-01-22

Reject